# Red, Gold and Green: Microbial Contribution of Rhodophyta and Other Algae to Green Turtle (*Chelonia mydas*) Gut Microbiome

**DOI:** 10.3390/microorganisms10101988

**Published:** 2022-10-08

**Authors:** Lucía Díaz-Abad, Natassia Bacco-Mannina, Fernando Miguel Madeira, Ester A. Serrao, Aissa Regalla, Ana R. Patrício, Pedro R. Frade

**Affiliations:** 1CCMAR—Centre of Marine Sciences, CIMAR, University of Algarve, 8005-139 Faro, Portugal; 2IMBRSea, International Master of Science in Marine Biological Resources, IMBRSea Universities Consortium, 9000 Ghent, Belgium; 3cE3c—Centre for Ecology, Evolution and Environmental Changes, CHANGE—Global Change and Sustainability Institute, Faculdade de Ciências da Universidade de Lisboa, 1749-016 Lisbon, Portugal; 4CIBIO/InBIO—Centro de Investigação em Biodiversidade e Recursos Genéticos, Universidade do Porto, 4485-661 Vairão, Portugal; 5IBAP—Instituto da Biodiversidade e das Áreas Protegidas Dr. Alfredo Simão da Silva, Bissau 1220, Guinea-Bissau; 6MARE—Marine and Environmental Sciences Centre, Ispa—Instituto Universitário, 1149-041 Lisbon, Portugal; 7Centre for Ecology & Conservation, College of Life and Environmental Sciences, University of Exeter, Penryn TR10 9FE, Cornwall, UK; 8Natural History Museum Vienna, 1010 Vienna, Austria

**Keywords:** 16S rRNA, microbiota, eDNA, *Chelonia mydas*, metabarcoding, macrophytes, sea turtles

## Abstract

The fitness of the endangered green sea turtle (*Chelonia mydas*) may be strongly affected by its gut microbiome, as microbes play important roles in host nutrition and health. This study aimed at establishing environmental microbial baselines that can be used to assess turtle health under altered future conditions. We characterized the microbiome associated with the gastrointestinal tract of green turtles from Guinea Bissau in different life stages and associated with their food items, using 16S rRNA metabarcoding. We found that the most abundant (% relative abundance) bacterial phyla across the gastrointestinal sections were Proteobacteria (68.1 ± 13.9% “amplicon sequence variants”, ASVs), Bacteroidetes (15.1 ± 10.1%) and Firmicutes (14.7 ± 21.7%). Additionally, we found the presence of two red algae bacterial indicator ASVs (the Alphaproteobacteria *Brucella pinnipedialis* with 75 ± 0% and a Gammaproteobacteria identified as methanotrophic endosymbiont of *Bathymodiolus*, with <1%) in cloacal compartments, along with six bacterial ASVs shared only between cloacal and local environmental red algae samples. We corroborate previous results demonstrating that green turtles fed on red algae (but, to a lower extent, also seagrass and brown algae), thus, acquiring microbial components that potentially aid them digest these food items. This study is a foundation for better understanding the microbial composition of sea turtle digestive tracts.

## 1. Introduction

Gut microbiomes offer essential health benefits to their vertebrate hosts [1] as vital contributors to host digestion and utilization of complex food particles and to the proliferation of their intestinal epithelium within gastrointestinal (GI) tracts [2,3]. Assessing the microbial communities present in the GI tract of vertebrates is becoming fundamental to increase knowledge on their health and biology [4] by offering an insight into its functions and dysfunctions [5] and, hence, aiding managers in the establishment of conservation and protection measures [6,7].

Previous studies have assessed the green sea turtle (*Chelonia mydas*) gut microbiomes [8,9,10,11]. These have found, for example, that microbiomes differ between live-caught stranded turtles [8], as well as over time for turtles kept in recovery centers (i.e., before and after rehabilitation, [10]), suggesting an important input from the surrounding environment. Green turtles, and marine turtles in general, lack post-oviposition parental care and throughout the period of early development, juveniles have no close interaction with conspecific adults [9]. Thus, environmental interactions and habitat and diet shifts greatly influence their developing gut microbiome [9], which, for green turtles, seems to change soon after their settlement in coastal waters [11]. The increase in turtle body size and environmental temperature [12], along with the consumption of turtle feces by juveniles [11], facilitate the acquisition of a bacterial community adapted to digest polysaccharides, which enables green turtles to implement an herbivorous diet soon after recruitment [11]. However, green turtles display high levels of regional variability in the degree of omnivory and relative importance of seagrasses and seaweeds in their diets [13]. Potential differences in their microbiota (between seagrass and macroalgae feeders) have been suggested, as the profiles of the fatty acids produced in the large intestine differ between green turtles feeding on either of these two sources [2,14]. Yet, more studies are needed to better understand variation in gut microbiome of green turtles after recruitment to neritic habitats. In parallel, little is known about the dependence of gut microbiota on the diet of turtles, which is known to be one of the principal factors influencing GI microbial communities within all vertebrates [15]. 

The digestive tract of adult green turtles relies on hindgut fermentation, using cellulolytic microbes to break down vegetal material in the cecum and colon [16,17]. Price et al. [9] found a high presence of the Proteobacteria phylum, and low occurrence of bacteria associated with the fermentation of structural polysaccharides, in cloacal samples from pelagic and recently settled green turtles. Bacteria in the genera *Clostridium*, *Peptoclostridium* and *Cellulosilyticum* have been found in high abundance in wild green turtles and are known to be able to assimilate some of the main polysaccharides found in seagrass and other plant fibers, such as cellulose, hemicellulose and xylan [18,19]. Additionally, the predominance of Firmicutes and Bacteroidetes phyla in cloacal swab samples of wild healthy green turtles from the Great Barrier Reef and Brazil [20], supports the importance of these bacteria in a healthy gut microbiome. Firmicutes are highly relevant in the fermentation of complex polysaccharides [21,22] and Bacteroidetes contribute to the initial assimilation of both simple and complex carbohydrates [23]. 

Like their animal counterparts, algae also harbor abundant and diverse microbial communities, as they have been coexisting with bacteria ever since their early evolution [24]. Seaweed species live in association with microbial communities [25,26,27] which use some of the monosaccharides constituting the cell wall of macroalgae, such as rhamnose, xylose, glucose, mannose and galactose [28], as a source of carbon and energy [29]. Bacterial communities, in return, produce plant growth-promoting substances, bioactive compounds and other effective molecules that are responsible for the development and growth of macroalgae [30]. These algae-associated microbial communities vary between host species [25], between individuals of the same species [31] and between seasons [25] and locations [27]. Nevertheless, a core microbiome appears to exist amongst macroalgal-associated bacteria, consisting of members of Gammaproteobacteria, Bacteroidetes (CFB group), Alphaproteobacteria, Firmicutes and Actinobacteria [32]. 

Guinea-Bissau hosts a major green turtle population, with most of the nesting concentrated in the southeast of the Bijagós archipelago, and foraging grounds spreading through this archipelago [33], with an important one around the westernmost islands [34]. Two recent studies found that juvenile green sea turtles in this foraging ground feed mostly on red algae [35,36], thus, it is likely that red algae microbiomes are present in the GI tract of these turtles. Species of red algae, such as *Laurencia dendroidea*, *Spyridia filamentosa* and the corallinaceae, *Jania* sp., are known to be part of the local benthic macrophyte composition of the Bijagós archipelago in Guinea-Bissau [35]. Studies on particular Rhodophyta species, such as *L. dendroidea*, and *Corallina officinalis*, have revealed that the microbiome of the first one is dominated by nitrogen-fixing Cyanobacteria and aerobic heterotrophic Proteobacteria [37], while the later one hosts a very diverse microbial community, dominated by Proteobacteria, Cyanobacteria, Bacteroidetes, Actinobacteria, Planctomycetes, Acidobacteria, Verrucomicrobia, Firmicutes and Chloroflexi phyla [38].

DNA metabarcoding has already been applied to assess sea turtle behavior [39] and diet composition [35,40,41]. Here, we assess the gut microbiome of green turtles (juveniles and hatchlings) from the Bijagós archipelago, Guinea-Bissau, by targeting the 16S rRNA gene. We characterize the microbial community residing on local environmental algae and seagrass specimens belonging to nine different species in order to evaluate the contribution of microbial members of potential food items, particularly of red algae, to the digestive microbiomes of green turtles. Moreover, we establish microbial baselines for different green turtle GI compartments and the diverse food item groups.

## 2. Materials and Methods

### 2.1. Study Sites

All samples used in this study came from two distinct locations in Guinea-Bissau, West Africa. Esophageal and cloacal swabs were collected from juveniles foraging in coastal waters around the islands of Unhocomo and Unhocomozinho (11°31′ N–16°40′ W, Figure 1a), in the westernmost limit of the Bijagós Archipelago, Guinea-Bissau, where important foraging grounds for juvenile green turtles exist [34,42]. Intestine and stomach samples were collected from hatchlings found dead at the nesting beach of Poilão Island (10°52′ N–15°43′ W, Figure 1b), which hosts one of the world’s largest green turtle nesting populations, producing over one million hatchlings every year [43]. Local potential food items (macroalgae and seagrass species) were collected in the nearshore waters around Unhocomo, Unhocomozinho and Poilão Islands. 

Sampling took place at the end of the wet season, which spans between May and November, peaking in August. We do not have the specific environmental conditions at the time but, overall, sea surface temperature average was 27.3 °C (ranging from 25.1 °C to 29.5 °C, [34]) and rainfall averages around 2000 mm per year [42].

### 2.2. Green Turtle Sampling 

Foraging turtles were captured with an entanglement net (800 m long, 4 m deep, 20 cm mesh size), deployed from a pirogue operated by local Bijagós fishermen, by enclosing their foraging site for periods of one hour (during which the net was constantly scanned for turtles). Seven juvenile green turtles (with sizes ranging from 36.5 cm to 48.1 cm of curved carapace length, with an average of 40.6 cm ± 4.5 cm SD) captured on 27 October 2019, were sampled for this study. For more information on the captured individuals, see Appendix A. Once captured, the individuals were securely positioned aboard the research vessel anchored next to the net and covered with a wet towel for processing and sampling, with the plastron on a flat surface and the front flippers held close to the body. For the esophageal sampling, the mouth of the turtle was kept open and the esophageal entrance sampled as deep as possible (making a gentle swirl 2–3 times with a sterile swab of 20 cm long) while avoiding any other parts of the mucosa. For cloacal samples, the swab was inserted 2 cm into the cloaca (2–3 swirls; prior cleaning of the cloacal opening with chlorhexidine and gauze). To reduce stress to the individuals, their handling was kept to the lowest (a maximum of 15 min per turtle). Swabs were kept in individual centrifuge tubes filled with 96% EtOH which were maintained at room temperature for two weeks, until stored at 2–5 °C for a week followed by 20 °C storage at the CCMAR laboratory. Dissections of the digestive tubes were performed on dead hatchlings found on the beach in Poilão Island. Stomachal and intestinal chambers were sampled with the swabbing method and samples were kept in microcentrifuge tubes with 96% EtOH. These samples were taken with two purposes: (1) to be used as methodological control, as the individuals from which they were taken did not ingest any food item nor had they been in contact with seawater [44,45]; (2) to investigate what the microbiome of the green sea turtles and the different compartments of their GI tract are composed of. New sterile latex gloves were used for each sample, each individual and each GI section.

### 2.3. Macrophyte Sampling 

Green turtle food items were collected in the same abovementioned period by free diving using latex gloves, and samples were transported in Ziploc bags filled with seawater. Algal species *Sargassum* sp. (n = 1), *Caulerpa* sp. (leaf, n = 2, and stolon, n = 2), *Hypnea* sp. (n = 1), *Ulva* sp. (n = 1), *Padina* sp. (n = 1), *Colpomenia* sp. (n = 1), *Dyctiota* sp. (n = 1) and *Rhodophyta* sp. (n = 1), and the seagrass *Halodule wrightii* (leaf, n = 2, and rhizome, n = 2) were sampled at Unhocomo. Samples consisted of a piece of 1 × 1 cm of each specimen (manipulation was done with gloves and EtOH was used to disinfect the materials used), which was placed into a microcentrifuge tube filled with buffer (DNA/RNA Shield from Zymo; 2 mL) and kept in dark conditions. Larger algal specimens were dried in the sun and stored in an herbarium for further taxonomic identification. 

### 2.4. DNA Extraction and Analysis of Metabarcoding Data 

Nucleic acid extraction was performed with the QIAGEN DNeasy PowerBiolfilm kit, following manufacturer’s instructions, to which the following steps were added: samples were placed individually in microcentrifuge tubes with 100 µL of Milli-Q water, for washing off EtOH and salt, before being pelleted at maximum speed for 3 min and transferred to the bead tube. Bead tubes were incubated at 65 °C for 30 min and then homogenized in a TissueLyzer for 3 min at 18,000 rpm. DNA extracts were sent on dry-ice to MR DNA (mrdnalab.com, accessed on 1 February 2021; Shallowater, TX, USA) and a 465-bp fragment of the 16S rRNA gene including V5–V7 variable regions, was amplified using the HotStarTaq Plus Master Mix Kit (Qiagen, Redwood City, CA, USA) and the primer set 799F-1193R (forward: 5′-AMCVGGATTAGATACCCBG-3′; reverse: 5′-ACGTCATCCCCACCTTCC-3′) which avoided chloroplast cross-amplification [46]. The cycling profile was as follows: 95 °C for 5 min, followed by 30 cycles of 95 °C for 30 s, 53 °C for 40 s and 72 °C for 1 min, after which a final elongation step at 72 °C for 10 min was performed. PCR products were checked in 2% agarose gels and samples were multiplexed using unique dual indices and were pooled together in equal proportions based on their molecular weight and DNA concentrations. Pooled samples were purified using calibrated Ampure XP beads and were then used to prepare an Illumina DNA library.

Samples were sequenced in an Illumina MiSeq platform with 2 × 300-bp paired-end approach and QIIME2 pipeline (Version 2019.7, https://qiime2.org, accessed on 1 February 2021) was used to analyze the demultiplexed reads as single nucleotide variants, as described by Bolyen et al. [47]. Using FASTqProcessor, forward and reverse primers were removed (http://www.mrdnalab.com/mrdnafreesoftware/fastq-processor.html, accessed on 1 February 2021) followed by quality control with the removal of chimeric sequences using the DADA2 algorithm (qiime dada2 denoise-paired; quality phred > 20; trimming = 170) [48], which was also used to group the sequences based on 100% sequence similarity, generating representative sequences, subsequently referred to as amplicon sequence variants (ASVs). For taxonomic assignment, a Naïve-Bayes classifier was trained on the SILVA v132 99% database [49]. To assign taxonomy, the trained classifier was applied to the representative sequences (qiime feature-classifier classify-sklearn). Plastid-derived sequence reads and singletons were removed from the dataset, which was then rarefied to an even sequencing depth of 8000 reads (minimum number of reads per sample).

All statistical analyses were performed in R software [50], separately for “turtle GI compartments” and for the “food item groups”. Alpha diversity was analyzed as richness (observed taxa, ASVs), along with Shannon and Chao1 diversity indexes [15]. The distribution of microbial communities across sampling groups (beta-diversity) was explored using Permutational Multivariate Analysis of Variance (PERMANOVA, ‘vegan package’ adonis2: [51]), Multivariate Homogeneity of Group Dispersion (PERMDISP, ‘vegan package’, betadisper: [51]) and Non-metric Multidimensional Scaling (NMDS, ‘phyloseq package’, [52,53]), which were all based on Bray Curtis dissimilarities. Graphs were created in R using ‘ggplot2′ [54] and ‘phyloseq’ [52] packages. PERMANOVA pairwise comparisons were corrected following the Bonferroni method [55].

The evaluation of the contribution of the microbiome associated with different food items to the gut microbiota of green turtles was addressed with three commonly used statistical approaches. Identification of unique and shared/cosmopolitan ASVs across the different sampling groups was based on present/absence data and accomplished with the ‘VennDiagram package’ [56]. Differential abundance analysis was performed on non-rarefied data using the ‘DESeq2 package’ [57,58] to recognize specific microbial ASVs causing significant community composition changes amongst pairwise comparisons of the different sampling groups. Lastly, to identify ASV diagnostics of particular sampling groups, an indicator species analysis, using the ‘indicspecies package’ [59,60], was applied.

Before the mentioned analyses, the food items were pooled into “food item groups” (from now onwards) according to higher taxonomic levels (Division: green (Chlorophyta) and red (Rhodophyta) algae; Class: brown algae (Phaeophyceae) and seagrass) in order to circumvent low replication. Diversity estimates are given as average number of ASVs ± SD (i.e., ASV richness), unless otherwise specified. For values based on one data point only, no variation is given.

## 3. Results

A total of 35 samples were sequenced, including cloaca and esophagus of seven juveniles (cloaca swabs = 7, esophagus swabs = 7), intestines and stomach from five hatchlings (intestine samples = 3, stomach samples = 3), as well as all of the diverse macrophytes (n = 15). From these, a total of 4,485,114 good quality reads (non-chimeric, Appendix A) (i.e., phred > 20) corresponding to 18.6% of the initial number of reads were generated (Appendix A). Samples had an average of 128,146 ± 70,916 reads, which clustered into 2311 unique ASVs across all samples combined (2303 ASVs after removal of chloroplast and mitochondria and 2108 ASVs after rarefaction and removal of singletons). One esophagus sample yielded a low number of reads (<8000) and was removed from all subsequent analyses.

Overall, the four turtle GI compartments had a very similar observed ASV richness (ANOVA, F_(3,15)_ = 0.02, *p* = 0.99), yielding 84 ASVs ± 2 ASVs per sample (Appendix A). Juveniles had a higher content of unique ASVs than hatchlings: 266 ASVs (26.8%), cloaca; 237 ASVs (23.8%), esophagus; 154 ASVs (15.5%), intestines; 156 ASVs (15.7%), stomach (Appendix A), and only 7 ASVs (0.70%) were shared across all of the digestive compartments together (Appendix A). In terms of Chao 1 (richness of ASVs) and Shannon diversity, stomach samples presented the highest values, while cloacal ones had the lowest ones (Appendix A), but these differences were not statistically significant (Chao 1: ANOVA, F_(3,15)_ = 0.02, *p* = 0.99; Shannon: ANOVA, F_(3,15)_ = 0.38, *p* = 0.77). Observed richness, with an overall average of 115 ± 11 ASVs per sample (Appendix A) along with Chao 1 alpha diversity index (Appendix A) did not significantly differ among food item groups (Observed: ANOVA, F_(3,11)_ = 0.14, *p* = 0.93; Chao 1: ANOVA, F_(3,11)_ = 0.09, *p* = 0.97), while Shannon index did differ among food item groups (Shannon: ANOVA, F_(3,11)_ = 3.88, *p* = 0.04; Appendix A). Across all food item groups, 28 ASVs (2.4%) were shared (Appendix A), while food item groups and turtle digestive compartments together had 13 ASVs (0.62% out of the total of 2108 ASVs) in common (Appendix A). The greatest number of unique ASVs for a particular food item was found in green algae (328; 27.5%), whereas red algae showed the lowest number of unique ASVs (128; 10.7%) (Appendix A).

The microbial community structure of turtle samples (all GI compartments pooled) was significantly different from that of pooled food item groups (PERMANOVA, F_(1,32)_ = 2.85, *p* = 1 × 10^−4^), as represented by NMDS (Figure 2). Multivariate dispersion did not differ significantly between food item groups and turtle compartments (PERMDISP, F_(1,32)_ = 0.03, *p* = 0.86). Microbial communities differed significantly among turtle GI compartments (PERMANOVA: F_(3,15)_ = 2.21, *p* = 1 × 10^−4^), perhaps due to a significant difference in the multivariate dispersion of the microbial community among compartments (PERMDISP, F_(3,15)_ = 20.71, *p* = 1.37 × 10^−5^, Appendix A). On the other hand, microbial communities did not appear to be significantly structured according to the food item group (PERMANOVA, F_(3,11)_ = 1.26, *p* = 0.08, Appendix A) and there was no difference in the multivariate dispersion between them (PERMDISP, F_(3,11)_ = 0.88, *p* = 0.48). Moreover, PERMANOVA pairwise comparisons (after Bonferroni correction) revealed no significant differences in community structure amongst the different food groups (*p* > 0.05 for all comparisons).

PERMANOVA pairwise comparisons (after Bonferroni correction) across all of the different sample types (turtle GI compartments and food item groups) revealed that significant differences in the microbial community structure only happened between juvenile samples (esophagus and cloaca) and green algae, while the rest of the GI compartments and food item groups had no significant differences in their microbial community compositions (Appendix A).

Microbial community composition varied across all sample types; nevertheless, there were two bacterial phyla present in all samples, which happened to be the most abundant ones: Proteobacteria (67.7 ± 10.3%) and Bacteroidetes (14.4 ± 5.9%) (Appendix A). A single (undetermined) family was present across all sample types and belonged to the Gammaproteobacteria class, being the third most abundant family (14.2 ± 13.8%), followed by Rhodobacteraceae (10.8 ± 9.8%), which were only absent from esophageal samples.

More specifically, microbial families across green turtle GI compartments showed different compositions and relative abundances (Figure 3A) but three phyla were dominant in all of the digestive sections: Proteobacteria (68.1 ± 13.9%), Bacteroidetes (15.1 ± 10.1%) and Firmicutes (14.7 ± 21.7%). Gammaproteobacteria_undet was the dominant family, present across all GI sections (56.8 ± 13.4%), either as first or second in order of abundance. It was the most abundant family in juvenile samples (38.95 ± 17.41%), and the second one in hatchlings (7.48 ± 10.38%). The family Enterobacteriaceae was present across all of the green turtle GI compartments, except in the intestines of hatchlings, and Rhodobacteraceae and Flavobacteriaceae families were only absent in the esophagus (Figure 3A).

Only two phyla were present within all food item groups, Proteobacteria (85.6 ± 13.4%), and Bacteroidetes (7.6 ± 4.1%) (Figure 3B), while four families showed up in all of the food item groups in high abundance, except in green algae: Gammaproteobacteria_undet (22.3 ± 18.5%), Rhodobacteraceae (18.5 ± 15.1%), Alteromonadaceae (8.6 ± 14.9%) and Flavobacteriaceae (7.3 ± 7.6%) (Figure 3B). Similar to the green turtle GI compartments, the Gammaprotecobacteria_undet family was present across all food item groups, being the second dominant one in red algae (2.5 ± 8.2%), while Rhodobacteraceae was the principal one in this food item group (5.4 ± 10.4%). Green algae were the food item group that presented the most distinct community composition (Figure 3B), showing the four above mentioned families, but in low abundance (<4%), as compared to the other food item groups.

Diagnostic ASVs for the different sample types were assessed by the INDVAL analysis. There were 24 indicator bacterial ASVs in the esophagus, six in the stomach and three in the intestines in green turtle GI compartments,. Besides, there was one common indicator bacterial ASV (Family: Gammaproteobacteria_undet.) for esophagus and stomach together (Appendix A). From all food item groups, only two of them were signalized as having indicators, with a total of 14 indicator ASVs: eight in brown algae and six in red algae (Appendix A). The INDVAL analysis, run for all of the sample types together (i.e., all food item groups and turtle GI compartments, Figure 4), gave the same result as the analysis done only on the turtle compartments, except for the absence of the indicator ASV for stomach and esophagus together (Appendix A).

Microbial families causing significant abundance differences between the distinct sample types were identified with the differential abundance analysis (DESeq2). Each green turtle GI tract compartment presented its own uniquely abundant families, while almost all food item groups (except red algae) showed a common dominant bacterial phylum, Planctomycetes, responsible for most statistical differences (see Appendix A for more details). Implementation of DESeq2 between the esophagus and the rest of the sample types revealed that Cardiobacteriaceae and Thiotrichaceae bacterial families were the ones displaying most significant differences in the microbial community composition, while Nannocystaceae and Lachnospiraceea families were the ones causing the differences when comparing cloacal and intestinal samples, respectively, to the rest of the sample types. Stomachs (of hatchlings) had significant differences in abundance in comparison to the rest of the groups, due to the high abundance of Enterobacteriaceae and Peptostreptococcaceae families. Red algae showed a high presence of Gammaproteobacteria_undet and Flavobacteriaceae families.

## 4. Discussion

Despite the extensive information that is now available on the trophic ecology of green turtles (i.e., [13,40,41,61,62]), insights on their gut microbial community composition are still scarce (but see [9,63,64]). Here, we explored the microbial community composition from four GI tract compartments of green sea turtles (Appendix A) from Guinea-Bissau, and from their potential food items (Appendix A) using 16S rRNA metabarcoding, while, at the same time, establishing microbial baselines for the different green turtle GI compartments and the diverse food item groups. Since juvenile green turtles from Unhocomo and Unhocomozinho, in Guinea-Bissau, feed mainly on red algae [35], we further investigated the influence of the microbial composition of this food item on that of the host gut.

### 4.1. Turtle GI Microbial Baselines

Our findings reflected the already known change in gut microbial community composition taking place when green turtle hatchlings start to feed, as they experience a strong environmental influence from their diet and from seawater on their bacterial communities [65]. Moreover, Gammaproteobacteria_undet seemed to be an important undetermined bacterial family within the Gammaproteobacteria class in the gut of the green turtle. It occurred as the dominant microbial family across juvenile samples and the second most abundant one in hatchlings and, this bacterial class, has also been found in abundance in stranded green turtles from the Great Barrier Reef [8].

The indicator ASVs found in hatchlings are potentially part of the core microbiome of green turtles, as hatchlings have not yet had any contact with food nor seawater [44,45]. In the same line, the seven ASVs shared across all of the GI compartments together (hatchlings plus juveniles) were identified as bacterial families that do not often occur in seawater, which brought us to also consider them as potential members of the core microbiome of the turtles and possibly, maternally inherited.

The scarce 13 ASVs shared across all sample types corroborated the visual NMDS results where food item groups overlapped with turtle compartments, and were identified as microbial families typically found in the marine environment (Marinicella_undet; [66], Erythrobacter_undet; [67], Lewinella_undet; [68]) leading to the conclusion that these ASVs present across all sample types came from microbial families typically associated with seawater.

The esophagus was the GI compartment that presented the most homogeneous microbial composition, showing the greatest amount of indicator ASVs (Appendix A), many of which corresponded to bacteria commonly found in the marine environment (Gammaproteobacteria_undet, Bacteroidia_undet, Mariniflilum_undet, Corynebacterium1_undet, Bdellovibrio_undet, Nannocystaceae_undet, Hyphomicrobiaceae_undet, Desulfatiferula_berrensis, Muricauda_undet [27,69,70,71]. Since sea turtles regularly ingest bacteria associated with seawater, the variety of microbes residing in the mucosa of their esophagus can be altered by transient bacterial communities coming from the external environment [65]. As the esophagus is one of the first GI sections to be in contact with the marine environment, it serves as an entrance spot for (gut) microbes, hosting the highest diversity, which becomes reduced as these microbes move through the GI tract [65].

Hindgut microbial communities dominated by Proteobacteria, Firmicutes, Bacteroidetes and Verrucomicrobia, similar to our results, have also been found in other marine herbivorous organisms, such as in the rabbitfish, *Siganus fuscescens* [15]. Proteobacteria, Firmicutes and Bacteroidetes are also the dominant bacterial phyla in the gut of green sea turtles from different parts of the world [8,9,10,11,20,72]. Post rehabilitation green turtles from Australia presented a dominance of Proteobacteria followed by Bacteroidetes and Firmicutes [10]. Gut resident microbes from juvenile green turtles from coastal areas of Brazil have shown a codominance between Bacteroidetes and Firmicutes, and a high abundance of Proteobacteria [11].

Other sea turtle species have also been found to harbor the same dominant digestive microbiome phyla members that are typical of herbivore gut microbiota [20], such as the loggerhead (*Caretta caretta*, [11]), the olive ridley (*Lepidochelys olivacea*, [73]) and the hawksbill sea turtle (*Eretmochelys imbricata*, [74]), suggesting that despite being mostly carnivorous (feeding on sponges, soft corals and cnidarians), the loggerhead and the olive ridley might have the ability to digest algae and seagrass [75,76]. The occurrence of Proteobacteria in the GI tract of vertebrates has been suggested as an indication of disease and dysbiosis [77], but also as a possible response to stress associated with anthropogenic factors [11]. Bacteria within the phylum Bacteroidetes are common members of gut microbiota in many vertebrates, including healthy sea turtles [10,78]. Firmicutes are amongst the main gut microorganisms of sea turtles [78] and their presence across all GI compartments might be explained by their ability to break down plant-derived polysaccharides, such as cellulose, hemicellulose and xylan [10]. The Bacteroidetes and Firmicutes use nutrients from the food items consumed by the turtles [79,80], starting this activity in the caecum, while little digestion and absorption take place in the stomach [2]. Accordingly, we found that members of the phylum Firmicutes, especially the Lachnospiraceae family, had a high abundant presence in the intestines of hatchlings. Corroborating our findings, Ahasan et al. [65] demonstrated that Firmicutes are significantly dominant in green turtle stomachs and large intestines, which was further supported by analyses on fecal samples [8,9,11].

### 4.2. Microbial Contribution of Food Items to Green Turtle GI Microbiome

Since juvenile green turtles from Unhocomo and Unhocomozinho foraging grounds, in Guinea-Bissau, eat red algae [35], and our results revealed that microbial communities of red algae were not significantly different from the ones in the green turtle GI tract compartments, we further focused on the microbiome of this food item group to evaluate its potential contribution to the digestive microbiome of the green turtles. Nonetheless, as the microbial communities of brown algae and seagrass were neither significantly different from the ones in the green turtle GI tract compartments, we did not discard the possibility of these food item groups influencing the gut microbiome as well, as these green turtles also feed on seagrass, and, in a lower degree, on brown algae [35].

Bacterial phyla, such as Firmicutes, Planctomycetes, Verrucomicrobia and Cyanobacteria, along with Bacteroidetes and Proteobacteria [25,81], have been recognized as frequent colonizers of red algae [82], which was also supported by our findings (Appendix A). One undetermined Gammaproteobacteria microbial family was the second most abundant one in this algae food item group, and it was one of the families causing significant microbial community composition differences in the DESeq2 analysis; some of the bacterial indicator ASVs of red algae belonged to this undetermined family as well. Members of this same bacterial class have been isolated from red algae species, such as *Amphiroa anceps* and *Corallina officinalis* [83].

In contrast with the 14 microbial ASVs indicators of the food item groups, there were no indicator ASVs of food items when this analysis was run for food item groups and green turtle GI compartment samples together. This showed that these ASVs were present in the turtle microbiome probably as a result of green turtles eating those food items. From the six red algae indicator ASVs in the first analysis, only two of them (Alphaproteobacteria_undet, NCBI BLAST: *Brucella pinnipedialis*; and Gammaprotebacteria_undet, NCBI BLAST: Methanotrophic bacterial endosymbiont of *Bathymodiolus* sp.) were found in green turtle GI compartments, in both cloaca and stomach (Figure 5A). While the indicator ASV identified as Alphaproteobacteria_undet had a 75 ± 0% abundance in cloacal samples and 25% in the stomach, the Gammaprotebacteria_undet indicator ASV abundance in both compartments was very low (<1%).

The presence of these ASV microbial indicators of red algae in the cloaca (Figure 5A), as well as of shared bacterial ASVs between cloaca and red algae (Figure 5B), was most likely due to their acquisition through the ingestion of red algae [35]. In fact, two of the green turtle individuals that presented the two red algae indicators (Animal J6 and J7, Figure 5A) along with animals J2, J5, J6 and J7 that shared microbial ASVs (in their cloaca) with red algae (Figure 5B), were the same individuals for which a parallel study on diet analysis identified red algae as the main diet items [35].

From the six microbial ASVs shared between green turtle GI compartments and red algae, all were present in cloacal samples (in different abundances and individuals), and two ASVs were found in one hatchling (Animal H2, intestines, Figure 5B). The same hatchling individual harbored a red algae indicator ASV in its stomach, which could hypothetically be due to contamination, but the possibility of vertical transmission should not be discarded [84]. As not all of the cloacal samples that presented shared ASVs with red algae (Figure 5B) and red algal indicator ASVs (Figure 5A) belonged to individuals found to have eaten red algae [35], we suggest the possibility that, in addition to recent acquisition from the diet [65], red algae indicators may actually be integrated in the core microbiome of the turtles and passed on via vertical transmission. This is an open research question that needs to be addressed in the future for marine turtles, as for freshwater turtles, such as the yellow-spotted Amazon River turtle (*Podocnemis unifilis*), the gut microbiome of which is known to be partially determined by vertical transmission [85].

Apart from cellulose and galactans, the cell walls of red algae have high protein content [86,87]. Some species of Alphaproteobacteria have the ability to hydrolize cellulose and lignin [88,89], while members of Gammaproteobacteria are capable of hydrocarbon utilization [30]. Marine Bacteroidetes are degraders of particulate matter, of proteins principally, as they can live attached to particles and have the capacity to degrade polymers, including peptidases, glycoside hydrolases, glycosyl transferases and adhesion proteins [90]. Particularly, Flavobacteriaceae bacteria are capable of degrading the main components of red algae cell walls [86], complex carbohydrates and proteins [90,91,92], due to the diversity of genes encoding their polymer-degrading enzymes [93]. Presence of the described taxa (Alphaproteobacteria, Gammaproteobacteria, Bacteroidetes, Flavobacteriaceae) in the cloaca of green turtles, if acquired by vertical transmission after initial incorporation from red algae microbiome, could indicate that they are important contributors to the digestion and degradation of the turtle food items, especially of red algae. Further research addressing functions other than assistance to digestion by the microbiome of food items could give more information on the health status and immune systems of the sea turtles [5].

### 4.3. Limitations and Recommendations

Taxonomic assignment of the microbial communities associated with the GI tract of the green turtles and of their potential food items was done using the best available reference library (SILVA, [49], see methods section). Nevertheless, the microbial identification could improve with updated microbial databases. Such a high proportion of unclassified reads, as found in other marine vertebrates [21,94], implies a significant presence of novel/undescribed bacteria, suggesting an important role for future research in the identification of unclassified bacterial within the GI tract of green turtles [8]. Moreover, we do not know if the sampled green turtle juveniles were feeding or not at the time of capture, and, thus, the analyzed microbial communities could be underrepresented relative to what would be expected in feeding animals [72]. On another line, cloacal samples may be unreliable indicators of digestive system microbiome due to contamination of external surfaces by seawater bacteria [20]. Additionally, we note that the replication used to characterize the microbiome of the different food item groups was relatively low and could have missed some of their rarest members. Further investigation, considering temporal and seasonal variations and during feeding events in a larger sample size, is suggested, to determine a more precise digestive baseline for green turtle microbial composition and for the evaluation of the role of diet on the gut microbiome. Finally, the real function of the taxa here revealed must be investigated in the future using transcriptomics on the active microbial fraction.

## 5. Conclusions

In line with our initial objectives, through the implementation of 16S rRNA metabarcoding, we revealed that green turtles experience a shift in gut microbial community composition when developing from hatchlings that just emerged from the nest, which likely include only core microbiome, to juveniles foraging in coastal waters, which, apart from the core microbiome, include transient microbial communities likely influenced by their environment. We suggest that the red algal microbiome present in the cloaca could potentially be part of the core microbiome of the turtles, but we cannot discard the possibility that it is a transient microbiota controlled by the diet. Our study represents the first detailed research of green turtle gut microbial communities in green turtles from Guinea Bissau. We hope to establish a baseline for a better understanding of the microbial composition of green sea turtle GI tracts and their feeding behavior, guiding the way to the future use of microbes as indicators for the quality of foraging grounds.

## Figures and Tables

**Figure 1 microorganisms-10-01988-f001:**
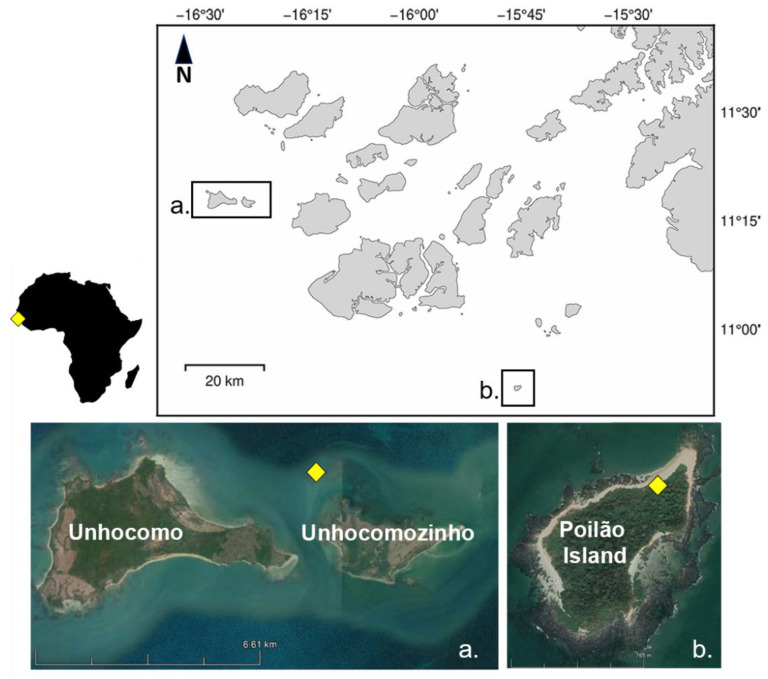
Map of study sites in the Bijagós Archipelago, Guinea-Bissau: (**a**) Unhocomo and Unhocomozinho islands; (**b**) Poilão, island.

**Figure 2 microorganisms-10-01988-f002:**
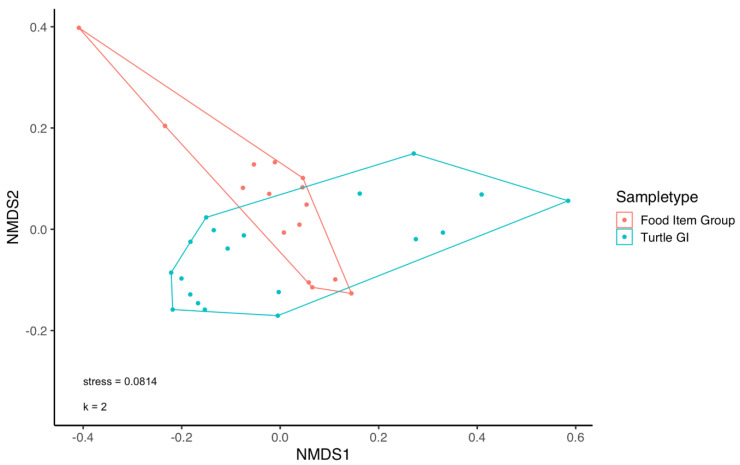
Microbial community structure organized according to the origin of samples collected for this study: green turtle gastro-intestinal (GI) compartments and green turtle putative food item groups (macroalgae and seagrass species). NMDS plot based on Bray-Curtis.

**Figure 3 microorganisms-10-01988-f003:**
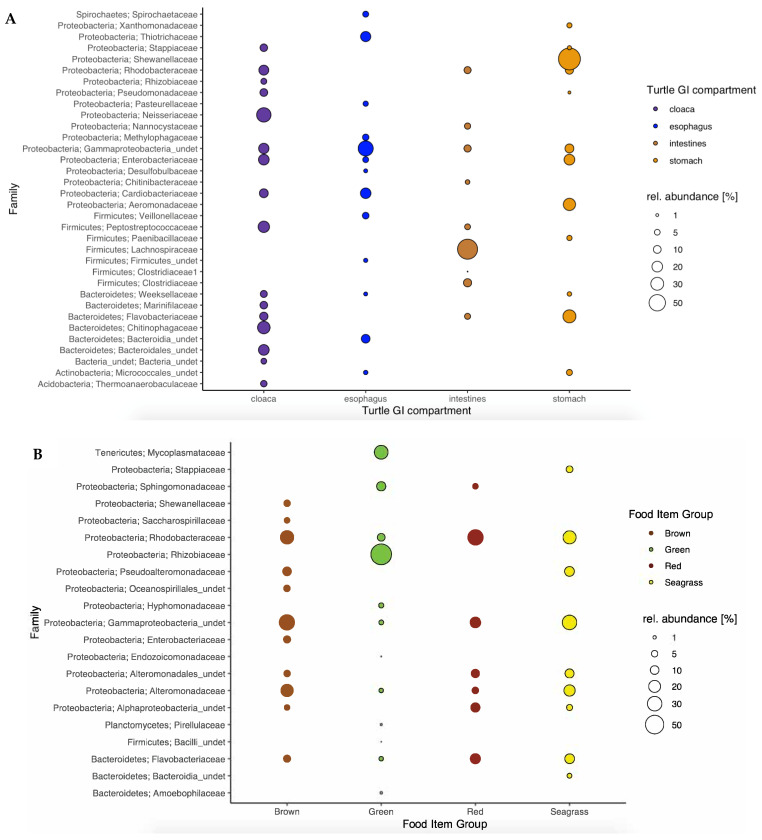
Microbial community composition at the family level (*Y* axis, Phylum; Family) of four green turtle gastro-intestinal (GI) compartments (**A**) and of different putative food item groups (**B**). Average of the abundance of the families across the samples of each GI compartment/food item group is represented as relative abundance. Abundance values <1% are not represented.

**Figure 4 microorganisms-10-01988-f004:**
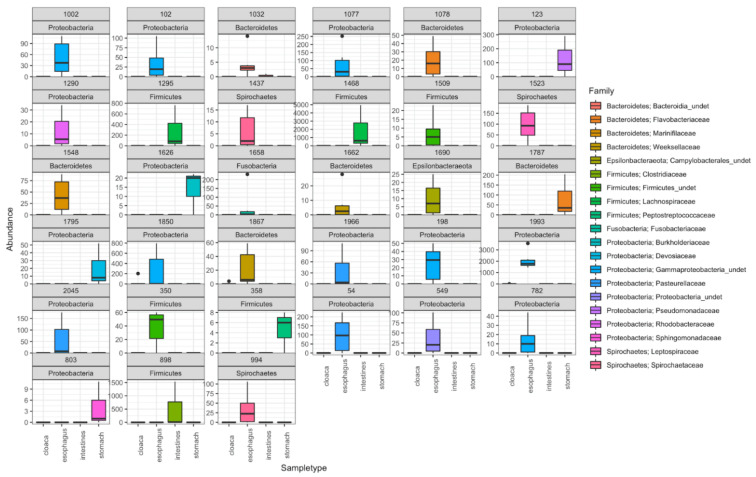
Indicator species analysis for green turtle gastro-intestinal (GI) compartments and for green turtle putative food item groups (seagrass and macroalgae species). The number of indicator ASVs and phylum affiliations appear at the top of each box plot. The corresponding family of each indicator ASV is listed in the legend. Each boxplot corresponds to one indicator and the compartment it appears on is labelled in the *X* axis. Abundance of the indicator in the expressed sample type is shown in the *Y* axis.

**Figure 5 microorganisms-10-01988-f005:**
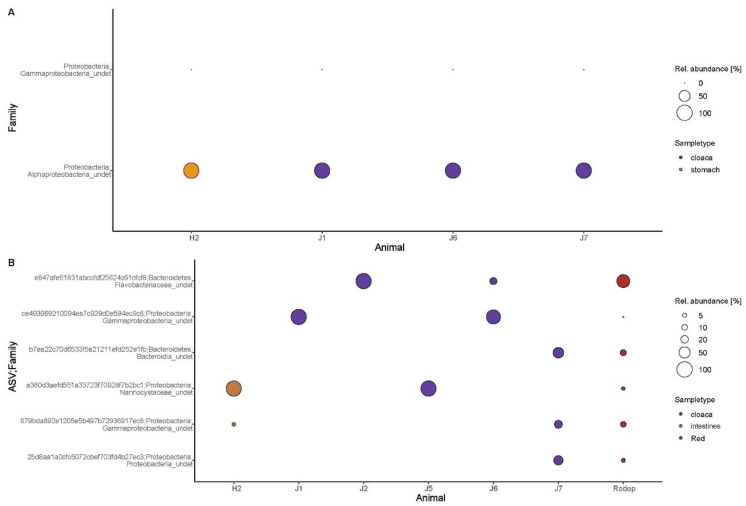
(**A**) Red algae indicator ASVs found in two of the green turtle GI compartments (cloaca: juveniles; stomach: hatchlings). Only two indicator ASVs (Phylum; Family level, *Y* axis) of red algae were found in only two of the green turtle GI sections: in cloaca and stomach. Presence of each ASV in each turtle individual (Animal) is represented as a circle in relative abundance. Indicator Gammaproteobacteria_undet rel. abundance was <1%. (**B**) Shared bacterial ASVs between red algae and turtle compartments (cloaca: juveniles; intestines: hatchlings). The six shared ASVs and the family level they are identified with are represented in the *Y* axis. The Animal code in the *X* axis in both A&B figures, corresponds to the individual ID they were sampled from. Presence of each ASV in each individual is represented as a circle in relative abundance.

## Data Availability

Demultiplexed sequences and metadata are available from the NCBI Sequence Read Archives (SRA) under BioProject Accession number PRJNA781615.

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
