# Peer review of "Red, Gold and Green: Microbial Contribution of Rhodophyta and Other Algae to Green Turtle (Chelonia mydas) Gut Microbiome"

_microorganisms, 2022, doi:10.3390/microorganisms10101988_

Round 1

Reviewer 1 Report

This is a nice study dealing with microbial associations between food sources and intestines of green turtle which has been studied only marginally. The study is comprehensive and well written, despite the fact that the paper is only descriptive and the results are based on quite small sampling effort. I think that it can serve as an important source of information for future studies and it can significantly contribute to the body of literature dealing with ecology of marine turtles. Also, the knowledge about interactions with microbial trophic level can also improve conservation efforts of this endangered species. I have a only few suggestions and comments to be addressed (see below):

Introduction

·        Lines 66-68: The sentence dedicated to importance of NGS is nowadays a cliche and thus redundant, please remove the sentence.

·        Line 91: The examples of the core taxa are too general , please remove them or modify the sentence in the following manner: … consisting of the members of Gammaproteobacteria, Bacteroidetes ….

Methods

·        Macrophyte sampling – is it enough to sample algae only in one biological replication per species? Together with only small part of plant tissue excised (1cm2), there could be the space for underestimation of real microbial diversity. I find this methodological issue as a serious deficiency of presented study. I think that low number of shared ASV between food item groups and turtle digestive compartments (only 13 ASVs, 0.62%) and other related results could be an artifact of insufficient sampling effort.

·        Moreover, there is no taxonomic or DNA identification of plant composition directly from the guts of sampled individuals, which leaves room for the question of whether specific individuals really consumed the mentioned plants (despite its evidence based on previous studies or parallel study of gut content). These constraints are only partially mentioned in the Discussion and definitelly deserve more attention in section 4.3 (Limitations of the study).

·        Line 176 – MRDNA – please provide the full name of the workplace

·        Line 178 – please provide details of or at least the link to used cycling profile

·        Line 179 – I understand that the second part of the library was prepared as a service at different workplace and the authors probably do not know further details about PCR and ligation procedure. However, could you please try to provide as many information as possible (which ligation kit was used, how many PCR replicates per sample were done, what was the final concentration of library etc.)?

·        Line 180 – QIIME2 is not software package but rather the pipeline or platform, and DADA2 is an algorithm

·        Line 181 – Was QIIME2 used also for demultiplexing and quality filterering of raw reads (did you use the q2‐demux plugin)? Please add this information.

·        Line 188 – Please provide the full information which classifier was used together with naïve Bayes classifier (for example q2‐feature‐classifier classify-sklearn is commonly used)

·        Line 198 – please remove the left bracket before „adonis2“

·        Line 202 – please provide citation of Bonferroni correction

Data availability Statement

·        please provide the full accession number of your Bioproject

Supplementary material

·        please check if all the captions are correct. It also seems that video is not available for reviewers.

Results

·        Please move the first paragraph about turtle numbers and characteristics (together with link to S1 Table) to the M&M section 2.2 (turtle sampling). Similarly, the first sentence of the second paragraph (Lines 225–27) about total number of sampled and sequenced specimens will look better in the M&M section.

·        Line 221 – „in“ is redundant

·        Lines 230–31 – please provide the number and percentage of discarderd mitochondrial and chloroplast sequences

·        Line 244 – left bracket is missing

Discussion

·        Please unify the term 16S rRNA metabarcoding and DNA metabarcoding in the manuscript

·        The discussion only describes the potential function of microorganisms but real evidence of their function can only be provided by a comprehensive transcriptomics approach and/or manipulative experiments. I think that the limitations of the study should also include the sentence that the real function and importance of revealed taxa must be investigated and proved by transcriptomics of active microbial fraction.

Author Response

COMMENT

This is a nice study dealing with microbial associations between food sources and intestines of green turtle which has been studied only marginally. The study is comprehensive and well written, despite the fact that the paper is only descriptive and the results are based on quite small sampling effort. I think that it can serve as an important source of information for future studies and it can significantly contribute to the body of literature dealing with ecology of marine turtles. Also, the knowledge about interactions with microbial trophic level can also improve conservation efforts of this endangered species. I have a only few suggestions and comments to be addressed (see below):

REPLY:

We really appreciate the reviewer’s words. We agree our study will provide a baseline for future research in this topic and will contribute to the study of endangered species by demonstrating the use of a non-intrusive monitoring method for microbiome characterization. We also agree the manuscript needed improvement and we were happy to follow the guidance for improvement provided by the reviewer.

Below we provide a point-by-point overview of the changes these comments have led to.

COMMENT:

Lines 66-68: The sentence dedicated to importance of NGS is nowadays a cliche and thus redundant, please remove the sentence.

REPLY:

We appreciate the reviewer’s suggestion and we have now removed this sentence. Thus, reference number [18] was removed and all subsequent citations renumbered.

COMMENT:

Line 91: The examples of the core taxa are too general, please remove them or modify the sentence in the following manner: … consisting of the members of Gammaproteobacteria, Bacteroidetes.

REPLY:

We appreciate the reviewer’s comment and have now modified the sentence as suggested. 

COMMENT:

Macrophyte sampling – is it enough to sample algae only in one biological replication per species? Together with only small part of plant tissue excised (1cm2), there could be the space for underestimation of real microbial diversity. I find this methodological issue as a serious deficiency of presented study. I think that low number of shared ASV between food item groups and turtle digestive compartments (only 13 ASVs, 0.62%) and other related results could be an artifact of insufficient sampling effort.

REPLY:

We agree that one replicate is not enough to characterise the microbial community associated with a particular species. However, we do not refer to the microbiome of any particular species, and we do not analyse these species individually. What we did was to pool these different samples together (for example, as green or red algae), in order to have an idea about the genera microbiome of these broader taxonomic groups. We were in fact limited by financial resources, otherwise we would have surely sequenced a larger number of replicates per macrophyte species. We have added a sentence to the section “4.3 Limitations and recommendations” in the discussion referring to this low replication of the different food item groups.

COMMENT:
Moreover, there is no taxonomic or DNA identification of plant composition directly from the guts of sampled individuals, which leaves room for the question of whether specific individuals really consumed the mentioned plants (despite its evidence based on previous studies or parallel study of gut content). These constraints are only partially mentioned in the Discussion and definitelly deserve more attention in section 4.3 (Limitations of the study).

REPLY:

We appreciate the reviewer comment. In fact, we did do DNA identification of plant composition directly from the sampled individuals (using esophageal samples). These data are published in a separate study (Diaz-Abad et al. 2022). We did refer to this other study throughout the text, and we did establish links to it when needed to support our argumentation that the gut microbiome must be linked to the consumption of certain food items. See, for example this sentence: “ In fact, two of the green turtle individuals that presented the two red algae indicators (Animal J6 and J7, Figure 5A) along with animals J2, J5, J6 and J7 that shared microbial ASVs (in their cloaca) with red algae (Figure 5B), were the same individuals for which a parallel study on diet analysis identified red algae as main diet items”. Furthermore, a new study from our research group (Madeira et al. 2022) using water captures and survey dives to record habitat use and characteristics, as well as stable isotopes to infer diet, has shown that this turtle population consumed some of the mentioned species (several Rhodophyta, mostly Laurencia sp., Caulerpa, Ochrophyta: Tribe: Dictyoteae, and Halodule). We have now also included this other study in our literature list, and we hope this addition (and the above explanation) are convincing enough to the reviewer.

COMMENT:

Line 176 – MRDNA – please provide the full name of the workplace

REPLY:

We appreciate the reviewer’s point and have now added the requested information.

COMMENT:

Line 178 – please provide details of or at least the link to used cycling profile

REPLY:

We appreciate the reviewer’s comment and have now incorporated the request information.

COMMENT:

Line 179 – I understand that the second part of the library was prepared as a service at different workplace and the authors probably do not know further details about PCR and ligation procedure. However, could you please try to provide as many information as possible (which ligation kit was used, how many PCR replicates per sample were done, what was the final concentration of library etc.)?

REPLY:

We appreciate the reviewer’s comment and have now incorporated the request information.

COMMENT:

Line 180 – QIIME2 is not software package but rather the pipeline or platform, and DADA2 is an algorithm.

REPLY:

We appreciate the reviewer’s opinion and have incorporated this suggestion.

COMMENT:

 Line 181 – Was QIIME2 used also for demultiplexing and quality filterering of raw reads (did you use the q2‐demux plugin)? Please add this information.

REPLY:

The raw data was already provided as demultiplexed by the sequencing company, and was otherwise untouched. QIIME 2, nevertheless, was used for quality filtering the demultiplexed sequences. Specific information on this topic has been added.

COMMENT:

 Line 188 – Please provide the full information which classifier was used together with naïve Bayes classifier (for example q2‐feature‐classifier classify-sklearn is commonly used)

REPLY:

We agree and are grateful for the reviewer’s suggestion and have now specified the information for this topic

COMMENT:

Line 198 – please remove the left bracket before „adonis2“

REPLY:

Done.

COMMENT:

Line 202 – please provide citation of Bonferroni correction

REPLY:

We appreciate and agree with the reviewer´s suggestion. We have now incorporated the respective bibliography:

Armstrong, RA (2014).  When to use the Bonferroni correction. Ophthalmic Physiol Opt  34:  502– 508.

https://doi.org/10.1111/opo.12131

DATA AVAILABILITY STATEMENT

Please provide the full accession number of your Bioproject

REPLY:

Done as requested.

SUPPLEMENTARY MATERIAL

Please check if all the captions are correct. It also seems that video is not available for reviewers.

REPLY:

We have now verified that all captions from the SOM are correct. However, Line 548 says Video S1, but we did not provide any Video as a Supplementary Material. Moreover, the Supplementary Material section, does not mention all of the Supplementary Material we have submitted.

COMMENT

Please move the first paragraph about turtle numbers and characteristics (together with link to S1 Table) to the M&M section 2.2 (turtle sampling). Similarly, the first sentence of the second paragraph (Lines 225–27) about total number of sampled and sequenced specimens will look better in the M&M section. 

REPLY:

We agree partially and we have moved the first paragraph about turtle numbers and characteristics to section 2.2. However, we would like to keep the information on total samples sequenced in the beginning of the results section, as to allow fluidity when reading into the results section. 

COMMENT

Line 221 – „in“ is redundant

REPLY:

We apreciate the degree of detail in the reviewer’s comment and have now incorporated the suggestion.

COMMENT

Lines 230–31 – please provide the number and percentage of discarderd mitochondrial and chloroplast sequences

REPLY:

We apreciate the comment of the reviewer and have now included the requested information.

COMMENT

Please unify the term 16S rRNA metabarcoding and DNA metabarcoding in the manuscript.

REPLY:

We appreciate the reviewer´s suggestions and we have now unified the term 16S rRNA metabarcoding throughout the main text.  We have kept the term DNA metabarcoding when addressing generically to this methodology.

COMMENT

The discussion only describes the potential function of microorganisms but real evidence of their function can only be provided by a comprehensive transcriptomics approach and/or manipulative experiments. I think that the limitations of the study should also include the sentence that the real function and importance of revealed taxa must be investigated and proved by transcriptomics of active microbial fraction.

REPLY:

We appreciate the reviewer´s suggestion and have incorporated the requested sentence.   

Reviewer 2 Report

Review for the paper "Red, gold and green: microbial contribution of Rhodophyta and other algae to green turtle (Chelonia mydas) gut microbiome" by Lucia Diaz-Abad, Natassia Bacco-Mannina, Fernando Miguel Madeira, Ester A. Serrão, Aissa Regalla, Ana R. Patrício, Pedro R. Frade submitted to "Microorganisms".

General comment.

The paper was focused on the gut microbiome of the endangered green sea turtle (Chelonia mydas). Studies dealing with microbes of different animals are now considered to be very important because the microorganisms play a pivotal role in animal health and fitness. Using 16S rRNA metabarcoding the authors obtained baseline data regarding the composition and relative contribution of major microbial groups inhabiting the gastrointestinal tract of green turtles caught in Guinea Bissau. They revealed Proteobacteria, Bacteroidetes and Firmicutes to be the most numerous microorganisms occurring in the digestive tract of the green turtle. The manuscript is generally well written in terms of presentation. It is based on a comprehensive data set. Methods seem to be valid for this kind of the paper. The main results are illustrated with relevant figures. The Discussion provides a good interpretation of the main findings. In general, the article improves the knowledge about the microbiome of the green turtle inhabiting West Africa. After minor revision, this paper may be accepted for publication in "Microorganisms".

Specific remarks.

Section 2.1. Provide information on the period of the study (dry/rain, season etc.). Also, a short description of environmental conditions (temperature, rainfall, etc.) during the sampling period is needed for better understanding.

Section 2.3. Please, specify if the tissues (leaf, thallii, rhyzoma) were analysed separately or together.

L224. I strongly suggest the authors move Table S1 from the Supplementary Data to the body of the ms.

L562, 575, 580, 583, 585, 588, 603, 638, 651, 659, 677, 698, 701, 703. Chelonia mydas should be italicized.

L622. Zobellia galactanivorans should be italicized.

L640. Laurencia dendroidea should be italicized.

L642. Corallina officinalis should be italicized.

L656, 658, 729. Caretta caretta should be italicized.

L659. Dermochelys coriacea should be italicized.

L663. Acropora muricata should be italicized.

L705. Marinicella sediminis should be italicized.

L708. Erythrobacter should be italicized.

L710-711. Lewinella, Lewinella cohaerens, Lewinella nigricans, Lewinella persica, Lewinella lutea, Lewinella marina should be italicized.

L722. Lepidochelys olivacea should be italicized.

L723. Chelonia mydas agassizii should be italicized.

L726-727. Eretmochelys imbricate, Fusarium falciforme should be italicized.

L731. Eretmochelys imbricata should be italicized.

L746. Heliocidaris erythrogramma should be italicized.

Author Response

We greatly appreciate the reviewer’s words and endorsement, and we would like to thank also for the comments and level of detail in the revision provided.

We did our best to incorporate all suggestions by the reviewer, and below we provide a point-by-point overview of the changes these comments have led to. 

Section 2.1. Provide information on the period of the study (dry/rain, season etc.). Also, a short description of environmental conditions (temperature, rainfall, etc.) during the sampling period is needed for better understanding. 

We agree with the reviewer’s comment and we have now added information in order to address these reasonable concerns.

Section 2.3. Please, specify if the tissues (leaf, thallii, rhyzoma) were analysed separately or together.

They were analyzed separately and this is now explicitly mentioned in the section: “Caulerpa sp. (leaf, n=2, and stolon, n=2)”.

L224. I strongly suggest the authors move Table S1 from the Supplementary Data to the body of the ms.

We respectfully disagree because we do not think that Table S1 is crucial to understand the study. As additional information it is, we would like to keep Table S1 as supplementary information. However, if the editor considers that the change is needed, then we will move it to the main body.

L562, 575, 580, 583, 585, 588, 603, 638, 651, 659, 677, 698, 701, 703. Chelonia mydas should be italicized.

L622. Zobellia galactanivorans should be italicized.

L640. Laurencia dendroidea should be italicized.

L642. Corallina officinalis should be italicized.

L656, 658, 729. Caretta caretta should be italicized.

L659. Dermochelys coriacea should be italicized.

L663. Acropora muricata should be italicized.

Done as requested.

Round 2

Reviewer 1 Report

Thank to authors for their improvements and taking into account the vast majority of my suggestions. I have no further comments. However, the issues with missing and mislabeled Supplementary material must be solved.

Author Response

We are not sure what the reviewer means, but we think it has to do with the section Supplementary Materials at the end of the manuscript, which we did not know was the authors' responsibility. So, we have now added the correct information to it. Please see attached new version of the manuscript. We also note that the info on the left side of the first page is not correctly filled in, but we will leave it to the editorial office to sort that. Finally, we noticed a mistake in our revision: In fact, there were 5 hatchling individuals instead of 3, and we have also corrected this in the new version now uploaded.